# FastGCN: Fast Learning with Graph Convolutional Networks via Importance Sampling

**Jie Chen**[*], **Tengfei Ma**[*], **Cao Xiao**
IBM Research
chenjie@us.ibm.com, Tengfei.Ma1@ibm.com, cxiao@us.ibm.com

## Abstract

The graph convolutional networks (GCN) recently proposed by Kipf and Welling are an effective graph model for semi-supervised learning. This model, however, was originally designed to be learned with the presence of both training and test data. Moreover, the recursive neighborhood expansion across layers poses time and memory challenges for training with large, dense graphs. To relax the requirement of simultaneous availability of test data, we interpret graph convolutions as integral transforms of embedding functions under probability measures. Such an interpretation allows for the use of Monte Carlo approaches to consistently estimate the integrals, which in turn leads to a batched training scheme as we propose in this work—FastGCN. Enhanced with importance sampling, FastGCN not only is efficient for training but also generalizes well for inference. We show a comprehensive set of experiments to demonstrate its effectiveness compared with GCN and related models. In particular, training is orders of magnitude more efficient while predictions remain comparably accurate.

## 1 Introduction

Graphs are universal representations of pairwise relationship. Many real world data come naturally in the form of graphs; e.g., social networks, gene expression networks, and knowledge graphs. To improve the performance of graph-based learning tasks, such as node classification and link prediction, recently much effort is made to extend well-established network architectures, including recurrent neural networks (RNN) and convolutional neural networks (CNN), to graph data; see, e.g., Bruna et al. (2013); Duvenaud et al. (2015); Li et al. (2015); Jain et al. (2015); Henaff et al. (2015); Niepert et al. (2016); Kipf & Welling (2016a;b).

Whereas learning feature representations for graphs is an important subject among this effort, here, we focus on the feature representations for graph vertices. In this vein, the closest work that applies a convolution architecture is the graph convolutional network (GCN) (Kipf & Welling, 2016a;b). Borrowing the concept of a convolution filter for image pixels or a linear array of signals, GCN uses the connectivity structure of the graph as the filter to perform neighborhood mixing. The architecture may be elegantly summarized by the following expression:

$$H^{(l+1)} = \sigma(\hat{A}H^{(l)}W^{(l)}),$$

where $\hat{A}$ is some normalization of the graph adjacency matrix, $H^{(l)}$ contains the embedding (row-wise) of the graph vertices in the $l$th layer, $W^{(l)}$ is a parameter matrix, and $\sigma$ is nonlinearity.

As with many graph algorithms, the adjacency matrix encodes the pairwise relationship for both training and test data. The learning of the model as well as the embedding is performed for both data simultaneously, at least as the authors proposed. For many applications, however, test data may not be readily available, because the graph may be constantly expanding with new vertices (e.g. new members of a social network, new products to a recommender system, and new drugs for functionality tests). Such scenarios require an inductive scheme that learns a model from only a training set of vertices and that generalizes well to any augmentation of the graph.

---

[*]These two authors contribute equally.

A more severe challenge for GCN is that the recursive expansion of neighborhoods across layers incurs expensive computations in batched training. Particularly for dense graphs and powerlaw graphs, the expansion of the neighborhood for a single vertex quickly fills up a large portion of the graph. Then, a usual mini-batch training will involve a large amount of data for every batch, even with a small batch size. Hence, scalability is a pressing issue to resolve for GCN to be applicable to large, dense graphs.

To address both challenges, we propose to view graph convolutions from a different angle and interpret them as integral transforms of embedding functions under probability measures. Such a view provides a principled mechanism for inductive learning, starting from the formulation of the loss to the stochastic version of the gradient. Specifically, we interpret that graph vertices are iid samples of some probability distribution and write the loss and each convolution layer as integrals with respect to vertex embedding functions. Then, the integrals are evaluated through Monte Carlo approximation that defines the sample loss and the sample gradient. One may further alter the sampling distribution (as in importance sampling) to reduce the approximation variance.

The proposed approach, coined FastGCN, not only rids the reliance on the test data but also yields a controllable cost for per-batch computation. At the time of writing, we notice a newly published work GraphSAGE (Hamilton et al., 2017) that proposes also the use of sampling to reduce the computational footprint of GCN. Our sampling scheme is more economic, resulting in a substantial saving in the gradient computation, as will be analyzed in more detail in Section 3.3. Experimental results in Section 4 indicate that the per-batch computation of FastGCN is more than an order of magnitude faster than that of GraphSAGE, while classification accuracies are highly comparable.

## 2 RELATED WORK

Over the past few years, several graph-based convolution network models emerged for addressing applications of graph-structured data, such as the representation of molecules (Duvenaud et al., 2015). An important stream of work is built on spectral graph theory (Bruna et al., 2013; Henaff et al., 2015; Defferrard et al., 2016). They define parameterized filters in the spectral domain, inspired by graph Fourier transform. These approaches learn a feature representation for the whole graph and may be used for graph classification.

Another line of work learns embeddings for graph vertices, for which Goyal & Ferrara (2017) is a recent survey that covers comprehensively several categories of methods. A major category consists of factorization based algorithms that yield the embedding through matrix factorizations; see, e.g., Roweis & Saul (2000); Belkin & Niyogi (2001); Ahmed et al. (2013); Cao et al. (2015); Ou et al. (2016). These methods learn the representations of training and test data jointly. Another category is random walk based methods (Perozzi et al., 2014; Grover & Leskovec, 2016) that compute node representations through exploration of neighborhoods. LINE (Tang et al., 2015) is also such a technique that is motivated by the preservation of the first and second-order proximities. Meanwhile, there appear a few deep neural network architectures, which better capture the nonlinearity within graphs, such as SDNE (Wang et al., 2016). As motivated earlier, GCN (Kipf & Welling, 2016a) is the model on which our work is based.

The most relevant work to our approach is GraphSAGE (Hamilton et al., 2017), which learns node representations through aggregation of neighborhood information. One of the proposed aggregators employs the GCN architecture. The authors also acknowledge the memory bottleneck of GCN and hence propose an ad hoc sampling scheme to restrict the neighborhood size. Our sampling approach is based on a different and more principled formulation. The major distinction is that we sample vertices rather than neighbors. The resulting computational savings are analyzed in Section 3.3.

## 3 TRAINING AND INFERENCE THROUGH SAMPLING

One striking difference between GCN and many standard neural network architectures is the lack of independence in the sample loss. Training algorithms such as SGD and its batch generalization are designed based on the additive nature of the loss function with respect to independent data samples. For graphs, on the other hand, each vertex is convolved with all its neighbors and hence defining a sample gradient that is efficient to compute is beyond straightforward.

Concretely, consider the standard SGD scenario where the loss is the expectation of some function $g$ with respect to a data distribution $D$:

$$L = \mathrm{E}_{x \sim D}[g(W; x)].$$

Here, $W$ denotes the model parameter to be optimized. Of course, the data distribution is generally unknown and one instead minimizes the empirical loss through accessing $n$ iid samples $x_1, \ldots, x_n$:

$$L_{\mathrm{emp}} = \frac{1}{n} \sum_{i=1}^{n} g(W; x_i), \qquad x_i \sim D, \ \forall i.$$

In each step of SGD, the gradient is approximated by $\nabla g(W; x_i)$, an (assumed) unbiased sample of $\nabla L$. One may interpret that each gradient step makes progress toward the sample loss $g(W; x_i)$. The sample loss and the sample gradient involve only one single sample $x_i$.

For graphs, one may no longer leverage the independence and compute the sample gradient $\nabla g(W; x_i)$ by discarding the information of $i$'s neighboring vertices and their neighbors, recursively. We therefore seek an alternative formulation. In order to cast the learning problem under the same sampling framework, let us assume that there is a (possibly infinite) graph $G'$ with the vertex set $V'$ associated with a probability space $(V', F, P)$, such that for the given graph $G$, it is an induced subgraph of $G'$ and its vertices are iid samples of $V'$ according to the probability measure $P$. For the probability space, $V'$ serves as the sample space and $F$ may be any event space (e.g., the power set $F = 2^{V'}$). The probability measure $P$ defines a sampling distribution.

To resolve the problem of lack of independence caused by convolution, we interpret that each layer of the network defines an embedding function of the vertices (random variable) that are tied to the same probability measure but are independent. See Figure 1. Specifically, recall the architecture of GCN

$$\tilde{H}^{(l+1)} = \hat{A} H^{(l)} W^{(l)}, \quad H^{(l+1)} = \sigma(\tilde{H}^{(l+1)}), \quad l = 0, \ldots, M-1, \quad L = \frac{1}{n} \sum_{i=1}^{n} g(H^{(M)}(i, :)). \tag{1}$$

For the functional generalization, we write

$$\tilde{h}^{(l+1)}(v) = \int \hat{A}(v, u) h^{(l)}(u) W^{(l)} \, dP(u), \quad h^{(l+1)}(v) = \sigma(\tilde{h}^{(l+1)}(v)), \quad l = 0, \ldots, M-1, \tag{2}$$

$$L = \mathrm{E}_{v \sim P}[g(h^{(M)}(v))] = \int g(h^{(M)}(v)) \, dP(v). \tag{3}$$

Here, $u$ and $v$ are independent random variables, both of which have the same probability measure $P$. The function $h^{(l)}$ is interpreted as the embedding function from the $l$th layer. The embedding functions from two consecutive layers are related through convolution, expressed as an integral transform, where the kernel $\hat{A}(v, u)$ corresponds to the $(v, u)$ element of the matrix $\hat{A}$. The loss is the expectation of $g(h^{(M)})$ for the final embedding $h^{(M)}$. Note that the integrals are not the usual Riemann–Stieltjes integrals, because the variables $u$ and $v$ are graph vertices but not real numbers; however, this distinction is only a matter of formalism.

Writing GCN in the functional form allows for evaluating the integrals in the Monte Carlo manner, which leads to a batched training algorithm and also to a natural separation of training and test data, as in inductive learning. For each layer $l$, we use $t_l$ iid samples $u_1^{(l)}, \ldots, u_{t_l}^{(l)} \sim P$ to approximately evaluate the integral transform (2); that is,

$$\tilde{h}_{t_{l+1}}^{(l+1)}(v) := \frac{1}{t_l} \sum_{j=1}^{t_l} \hat{A}(v, u_j^{(l)}) h_{t_l}^{(l)}(u_j^{(l)}) W^{(l)}, \quad h_{t_{l+1}}^{(l+1)}(v) := \sigma(\tilde{h}_{t_{l+1}}^{(l+1)}(v)), \quad l = 0, \ldots, M-1,$$

with the convention $h_{t_0}^{(0)} \equiv h^{(0)}$. Then, the loss $L$ in (3) admits an estimator

$$L_{t_0, t_1, \ldots, t_M} := \frac{1}{t_M} \sum_{i=1}^{t_M} g(h_{t_M}^{(M)}(u_i^{(M)})).$$

The follow result establishes that the estimator is consistent. The proof is a recursive application of the law of large numbers and the continuous mapping theorem; it is given in the appendix.

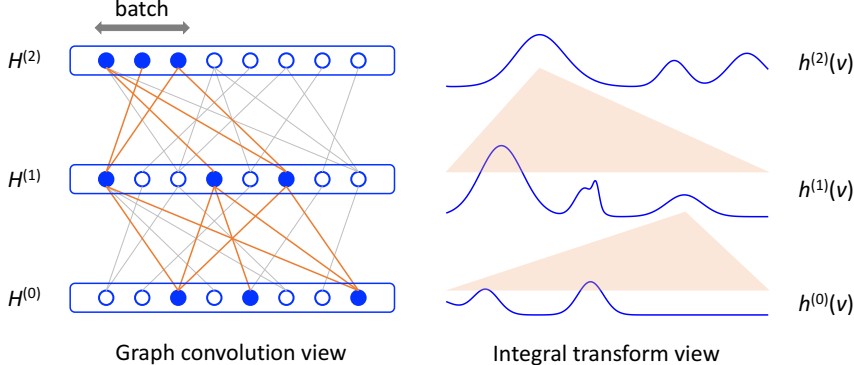

Figure 1: Two views of GCN. On the left (graph convolution view), each circle represents a graph vertex. On two consecutive rows, a circle $i$ is connected (in gray line) with circle $j$ if the two corresponding vertices in the graph are connected. A convolution layer uses the graph connectivity structure to mix the vertex features/embeddings. On the right (integral transform view), the embedding function in the next layer is an integral transform (illustrated by the orange fanout shape) of the one in the previous layer. For the proposed method, all integrals (including the loss function) are evaluated by using Monte Carlo sampling. Correspondingly in the graph view, vertices are subsampled in a bootstrapping manner in each layer to approximate the convolution. The sampled portions are collectively denoted by the solid blue circles and the orange lines.

**Theorem 1.** *If $g$ and $\sigma$ are continuous, then*

$$\lim_{t_0,t_1,\ldots,t_M \to \infty} L_{t_0,t_1,\ldots,t_M} = L \quad \text{with probability one}.$$

In practical use, we are given a graph whose vertices are already assumed to be samples. Hence, we will need bootstrapping to obtain a consistent estimate. In particular, for the network architecture (1), the output $H^{(M)}$ is split into batches as usual. We will still use $u_1^{(M)}, \ldots, u_{t_M}^{(M)}$ to denote a batch of vertices, which come from the given graph. For each batch, we sample (with replacement) uniformly each layer and obtain samples $u_i^{(l)}$, $i = 1, \ldots, t_l$, $l = 0, \ldots, M - 1$. Such a procedure is equivalent to uniformly sampling the rows of $H^{(l)}$ for each $l$. Then, we obtain the batch loss

$$L_{\text{batch}} = \frac{1}{t_M} \sum_{i=1}^{t_M} g(H^{(M)}(u_i^{(M)}, :)), \tag{4}$$

where, recursively,

$$H^{(l+1)}(v, :) = \sigma \left( \frac{n}{t_l} \sum_{j=1}^{t_l} \hat{A}(v, u_j^{(l)}) H^{(l)}(u_j^{(l)}, :) W^{(l)} \right), \quad l = 0, \ldots, M - 1. \tag{5}$$

Here, the $n$ inside the activation function $\sigma$ is the number of vertices in the given graph and is used to account for the normalization difference between the matrix form (1) and the integral form (2). The corresponding batch gradient may be straightforwardly obtained through applying the chain rule on each $H^{(l)}$. See Algorithm 1.

### 3.1 VARIANCE REDUCTION

As for any estimator, one is interested in improving its variance. Whereas computing the full variance is highly challenging because of nonlinearity in all the layers, it is possible to consider each single layer and aim at improving the variance of the embedding function before nonlinearity. Specifically, consider for the $l$th layer, the function $\tilde{h}_{t_{l+1}}^{(l+1)}(v)$ as an approximation to the convolution $\int \hat{A}(v, u) h_{t_l}^{(l)}(u) W^{(l)} \, dP(u)$. When taking $t_{l+1}$ samples $v = u_1^{(l+1)}, \ldots, u_{t_{l+1}}^{(l+1)}$, the sample average of $\tilde{h}_{t_{l+1}}^{(l+1)}(v)$ admits a variance that captures the deviation from the eventual loss contributed by this layer. Hence, we seek an improvement of this variance. Now that we consider each layer separately, we will do the following change of notation to keep the expressions less cumbersome:

---

**Algorithm 1** FastGCN batched training (one epoch)

---
1: **for** each batch **do**
2:     For each layer $l$, sample uniformly $t_l$ vertices $u_1^{(l)}, \ldots, u_{t_l}^{(l)}$
3:     **for** each layer $l$ **do**                                    ▷ Compute batch gradient $\nabla L_{\text{batch}}$
4:         If $v$ is sampled in the next layer,

$$\nabla \tilde{H}^{(l+1)}(v,:) \leftarrow \frac{n}{t_l} \sum_{j=1}^{t_l} \hat{A}(v, u_j^{(l)}) \nabla \left\{ H^{(l)}(u_j^{(l)}, :) W^{(l)} \right\}$$

5:     **end for**
6:     $W \leftarrow W - \eta \nabla L_{\text{batch}}$                                             ▷ SGD step
7: **end for**

---

|  | Function | Samples | Num. samples |
|---|---|---|---|
| Layer $l + 1$; random variable $v$ | $\tilde{h}_{t_{l+1}}^{(l+1)}(v) \rightarrow y(v)$ | $u_i^{(l+1)} \rightarrow v_i$ | $t_{l+1} \rightarrow s$ |
| Layer $l$; random variable $u$ | $h_{t_l}^{(l)}(u) W^{(l)} \rightarrow x(u)$ | $u_j^{(l)} \rightarrow u_j$ | $t_l \rightarrow t$ |

Under the joint distribution of $v$ and $u$, the aforementioned sample average is

$$G := \frac{1}{s} \sum_{i=1}^{s} y(v_i) = \frac{1}{s} \sum_{i=1}^{s} \left( \frac{1}{t} \sum_{j=1}^{t} \hat{A}(v_i, u_j) x(u_j) \right).$$

First, we have the following result.

**Proposition 2.** *The variance of $G$ admits*

$$\text{Var}\{G\} = R + \frac{1}{st} \iint \hat{A}(v, u)^2 x(u)^2 \, dP(u) \, dP(v), \tag{6}$$

*where*

$$R = \frac{1}{s} \left( 1 - \frac{1}{t} \right) \int e(v)^2 \, dP(v) - \frac{1}{s} \left( \int e(v) \, dP(v) \right)^2 \quad \text{and} \quad e(v) = \int \hat{A}(v, u) x(u) \, dP(u).$$

The variance (6) consists of two parts. The first part $R$ leaves little room for improvement, because the sampling in the $v$ space is not done in this layer. The second part (the double integral), on the other hand, depends on how the $u_j$'s in this layer are sampled. The current result (6) is the consequence of sampling $u_j$'s by using the probability measure $P$. One may perform importance sampling, altering the sampling distribution to reduce variance. Specifically, let $Q(u)$ be the new probability measure, where the $u_j$'s are drawn from. We hence define the new sample average approximation

$$y_Q(v) := \frac{1}{t} \sum_{j=1}^{t} \hat{A}(v, u_j) x(u_j) \left( \left. \frac{dP(u)}{dQ(u)} \right|_{u_j} \right), \qquad u_1, \ldots, u_t \sim Q,$$

and the quantity of interest

$$G_Q := \frac{1}{s} \sum_{i=1}^{s} y_Q(v_i) = \frac{1}{s} \sum_{i=1}^{s} \left( \frac{1}{t} \sum_{j=1}^{t} \hat{A}(v_i, u_j) x(u_j) \left( \left. \frac{dP(u)}{dQ(u)} \right|_{u_j} \right) \right).$$

Clearly, the expectation of $G_Q$ is the same as that of $G$, regardless of the new measure $Q$. The following result gives the optimal $Q$.

**Theorem 3.** *If*

$$dQ(u) = \frac{b(u)|x(u)| \, dP(u)}{\int b(u)|x(u)| \, dP(u)} \quad \text{where} \quad b(u) = \left[ \int \hat{A}(v, u)^2 \, dP(v) \right]^{\frac{1}{2}}, \tag{7}$$

*then the variance of $G_Q$ admits*

$$\text{Var}\{G_Q\} = R + \frac{1}{st} \left[ \int b(u)|x(u)| \, dP(u) \right]^2, \tag{8}$$

*where $R$ is defined in Proposition 2. The variance is minimum among all choices of $Q$.*

A drawback of defining the sampling distribution $Q$ in this manner is that it involves $|x(u)|$, which constantly changes during training. It corresponds to the product of the embedding matrix $H^{(l)}$ and the parameter matrix $W^{(l)}$. The parameter matrix is updated in every iteration; and the matrix product is expensive to compute. Hence, the cost of computing the optimal measure $Q$ is quite high.

As a compromise, we consider a different choice of $Q$, which involves only $b(u)$. The following proposition gives the precise definition. The resulting variance may or may not be smaller than (6). In practice, however, we find that it is almost always helpful.

**Proposition 4.** *If*

$$dQ(u) = \frac{b(u)^2 \, dP(u)}{\int b(u)^2 \, dP(u)}$$

*where $b(u)$ is defined in (7), then the variance of $G_Q$ admits*

$$\text{Var}\{G_Q\} = R + \frac{1}{st} \int b(u)^2 \, dP(u) \int x(u)^2 \, dP(u), \tag{9}$$

*where $R$ is defined in Proposition 2.*

With this choice of the probability measure $Q$, the ratio $dQ(u)/dP(u)$ is proportional to $b(u)^2$, which is simply the integral of $\hat{A}(v, u)^2$ with respect to $v$. In practical use, for the network architecture (1), we define a probability mass function for all the vertices in the given graph:

$$q(u) = \|\hat{A}(:, u)\|^2 / \sum_{u' \in V} \|\hat{A}(:, u')\|^2, \quad u \in V$$

and sample $t$ vertices $u_1, \ldots, u_t$ according to this distribution. From the expression of $q$, we see that it has no dependency on $l$; that is, the sampling distribution is the same for all layers. To summarize, the batch loss $L_{\text{batch}}$ in (4) now is recursively expanded as

$$H^{(l+1)}(v, :) = \sigma \left( \frac{1}{t_l} \sum_{j=1}^{t_l} \frac{\hat{A}(v, u_j^{(l)}) H^{(l)}(u_j^{(l)}, :) W^{(l)}}{q(u_j^{(l)})} \right), \quad u_j^{(l)} \sim q, \quad l = 0, \ldots, M - 1. \tag{10}$$

The major difference between (5) and (10) is that the former obtains samples uniformly whereas the latter according to $q$. Accordingly, the scaling inside the summation changes. The corresponding batch gradient may be straightforwardly obtained through applying the chain rule on each $H^{(l)}$. See Algorithm 2.

---

**Algorithm 2** FastGCN batched training (one epoch), improved version

---

1: For each vertex $u$, compute sampling probability $q(u) \propto \|\hat{A}(:, u)\|^2$
2: **for** each batch **do**
3:     For each layer $l$, sample $t_l$ vertices $u_1^{(l)}, \ldots, u_{t_l}^{(l)}$ according to distribution $q$
4:     **for** each layer $l$ **do**                                     ▷ Compute batch gradient $\nabla L_{\text{batch}}$
5:         If $v$ is sampled in the next layer,

$$\nabla \tilde{H}^{(l+1)}(v, :) \leftarrow \frac{1}{t_l} \sum_{j=1}^{t_l} \frac{\hat{A}(v, u_j^{(l)})}{q(u_j^{(l)})} \nabla \left\{ H^{(l)}(u_j^{(l)}, :) W^{(l)} \right\}$$

6:     **end for**
7:     $W \leftarrow W - \eta \nabla L_{\text{batch}}$                     ▷ SGD step
8: **end for**

---

## 3.2 INFERENCE

The sampling approach described in the preceding subsection clearly separates out test data from training. Such an approach is inductive, as opposed to transductive that is common for many graph algorithms. The essence is to cast the set of graph vertices as iid samples of a probability distribution, so that the learning algorithm may use the gradient of a consistent estimator of the loss to perform parameter update. Then, for inference, the embedding of a new vertex may be either computed by using the full GCN architecture (1), or approximated through sampling as is done in parameter learning. Generally, using the full architecture is more straightforward and easier to implement.

## 3.3 COMPARISON WITH GRAPHSAGE

GraphSAGE (Hamilton et al., 2017) is a newly proposed architecture for generating vertex embeddings through aggregating neighborhood information. It shares the same memory bottleneck with GCN, caused by recursive neighborhood expansion. To reduce the computational footprint, the authors propose restricting the immediate neighborhood size for each layer. Using our notation for the sample size, if one samples $t_l$ neighbors for each vertex in the $l$th layer, then the size of the expanded neighborhood is, in the worst case, the product of the $t_l$'s. On the other hand, FastGCN samples vertices rather than neighbors in each layer. Then, the total number of involved vertices is at most the sum of the $t_l$'s, rather than the product. See experimental results in Section 4 for the order-of-magnitude saving in actual computation time.

## 4 EXPERIMENTS

We follow the experiment setup in Kipf & Welling (2016a) and Hamilton et al. (2017) to demonstrate the effective use of FastGCN, comparing with the original GCN model as well as GraphSAGE, on the following benchmark tasks: (1) classifying research topics using the Cora citation data set (McCallum et al., 2000); (2) categorizing academic papers with the Pubmed database; and (3) predicting the community structure of a social network modeled with Reddit posts. These data sets are downloaded from the accompany websites of the aforementioned references. The graphs have increasingly more nodes and higher node degrees, representative of the large and dense setting under which our method is motivated. Statistics are summarized in Table 1. We adjusted the training/validation/test split of Cora and Pubmed to align with the supervised learning scenario. Specifically, all labels of the training examples are used for training, as opposed to only a small portion in the semi-supervised setting (Kipf & Welling, 2016a). Such a split is coherent with that of the other data set, Reddit, used in the work of GraphSAGE. Additional experiments using the original split of Cora and Pubmed are reported in the appendix.

Table 1: Dataset Statistics

| Dataset | Nodes | Edges | Classes | Features | Training/Validation/Test |
|---------|-------|-------|---------|----------|--------------------------|
| Cora | $2,708$ | $5,429$ | 7 | $1,433$ | $1,208/500/1,000$ |
| Pubmed | $19,717$ | $44,338$ | 3 | 500 | $18,217/500/1,000$ |
| Reddit | $232,965$ | $11,606,919$ | 41 | 602 | $152,410/23,699/55,334$ |

Implementation details are as following. All networks (including those under comparison) contain two layers as usual. The codes of GraphSAGE and GCN are downloaded from the accompany websites and the latter is adapted for FastGCN. Inference with FastGCN is done with the full GCN network, as mentioned in Section 3.2. Further details are contained in the appendix.

We first consider the use of sampling in FastGCN. The left part of Table 2 (columns under "Sampling") lists the time and classification accuracy as the number of samples increases. For illustration purpose, we equalize the sample size on both layers. Clearly, with more samples, the per-epoch training time increases, but the accuracy (as measured by using micro F1 scores) also improves generally.

An interesting observation is that given input features $H^{(0)}$, the product $\hat{A}H^{(0)}$ in the bottom layer does not change, which means that the chained expansion of the gradient with respect to $W^{(0)}$ in

Table 2: Benefit of precomputing $\hat{A}H^{(0)}$ for the input layer. Data set: Pubmed. Training time is in seconds, per-epoch (batch size 1024). Accuracy is measured by using micro F1 score.

| | Sampling | | Precompute | |
|---|---|---|---|---|
| $t_1$ | Time | F1 | Time | F1 |
| 5 | 0.737 | 0.859 | 0.139 | 0.849 |
| 10 | 0.755 | 0.863 | 0.141 | 0.870 |
| 25 | 0.760 | 0.873 | 0.144 | 0.879 |
| 50 | 0.774 | 0.864 | 0.142 | 0.880 |

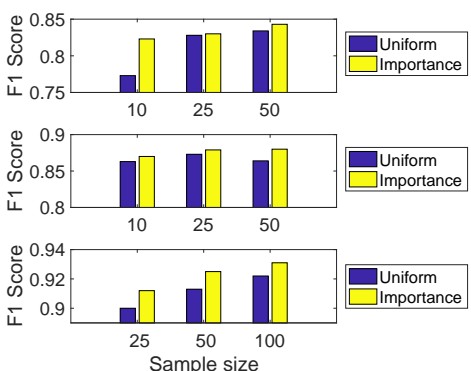

Figure 2: Prediction accuracy: uniform versus importance sampling. The three data sets from top to bottom are ordered the same as Table 1.

the last step is a constant throughout training. Hence, one may precompute the product rather than sampling this layer to gain efficiency. The compared results are listed on the right part of Table 2 (columns under "Precompute"). One sees that the training time substantially decreases while the accuracy is comparable. Hence, all the experiments that follow use precomputation.

Next, we compare the sampling approaches for FastGCN: uniform and importance sampling. Figure 2 summarizes the prediction accuracy under both approaches. It shows that importance sampling consistently yields higher accuracy than does uniform sampling. Since the altered sampling distribution (see Proposition 4 and Algorithm 2) is a compromise alternative of the optimal distribution that is impractical to use, this result suggests that the variance of the used sampling indeed is smaller than that of uniform sampling; i.e., the term (9) stays closer to (8) than does (6). A possible reason is that $b(u)$ correlates with $|x(u)|$. Hence, later experiments will apply importance sampling.

We now demonstrate that the proposed method is significantly faster than the original GCN as well as GraphSAGE, while maintaining comparable prediction performance. See Figure 3. The bar heights indicate the per-batch training time, in the log scale. One sees that GraphSAGE is a substantial improvement of GCN for large and dense graphs (e.g., Reddit), although for smaller ones (Cora and Pubmed), GCN trains faster. FastGCN is the fastest, with at least an order of magnitude improvement compared with the runner up (except for Cora), and approximately two orders of magnitude speed up compared with the slowest. Here, the training time of FastGCN is with respect to the sample size that achieves the best prediction accuracy. As seen from the table on the right, this accuracy is highly comparable with the best of the other two methods.

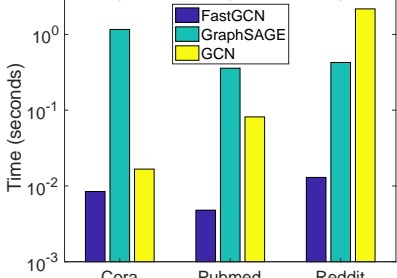

| Micro F1 Score | | | |
|---|---|---|---|
| | Cora | Pubmed | Reddit |
| FastGCN | 0.850 | 0.880 | 0.937 |
| GraphSAGE-GCN | 0.829 | 0.849 | 0.923 |
| GraphSAGE-mean | 0.822 | 0.888 | 0.946 |
| GCN (batched) | 0.851 | 0.867 | 0.930 |
| GCN (original) | 0.865 | 0.875 | NA |

Figure 3: Per-batch training time in seconds (left) and prediction accuracy (right). For timing, GraphSAGE refers to GraphSAGE-GCN in Hamilton et al. (2017). The timings of using other aggregators, such as GraphSAGE-mean, are similar. GCN refers to using batched learning, as opposed to the original version that is nonbatched; for more details of the implementation, see the appendix. The nonbatched version of GCN runs out of memory on the large graph Reddit. The sample sizes for FastGCN are 400, 100, and 400, respectively for the three data sets.

In the discussion period, the authors of GraphSAGE offered an improved implementation of their codes and alerted that GraphSAGE was better suited for massive graphs. The reason is that for small graphs, the sample size (recalling that it is the product across layers) is comparable to the graph size and hence improvement is marginal; moreover, sampling overhead might then adversely affect the timing. For fair comparison, the authors of GraphSAGE kept the sampling strategy but improved the implementation of their original codes by eliminating redundant calculations of the sampled nodes. Now the per-batch training time of GraphSAGE compares more favorably on the smallest graph Cora; see Table 3. Note that this implementation does not affect large graphs (e.g., Reddit) and our observation of orders of magnitude faster training remains valid.

Table 3: Further comparison of per-batch training time (in seconds) with new implementation of GraphSAGE for small graphs. The new implementation is in PyTorch whereas the rest are in TensorFlow.

|  | Cora | Pubmed | Reddit |
|---|---|---|---|
| FastGCN | 0.0084 | 0.0047 | 0.0129 |
| GraphSAGE-GCN (old impl) | 1.1630 | 0.3579 | 0.4260 |
| GraphSAGE-GCN (new impl) | 0.0380 | 0.3989 | NA |
| GCN (batched) | 0.0166 | 0.0815 | 2.1731 |

## 5 CONCLUSIONS

We have presented FastGCN, a fast improvement of the GCN model recently proposed by Kipf & Welling (2016a) for learning graph embeddings. It generalizes transductive training to an inductive manner and also addresses the memory bottleneck issue of GCN caused by recursive expansion of neighborhoods. The crucial ingredient is a sampling scheme in the reformulation of the loss and the gradient, well justified through an alternative view of graph convoluntions in the form of integral transforms of embedding functions. We have compared the proposed method with additionally GraphSAGE (Hamilton et al., 2017), a newly published work that also proposes using sampling to restrict the neighborhood size, although the two sampling schemes substantially differ in both algorithm and cost. Experimental results indicate that our approach is orders of magnitude faster than GCN and GraphSAGE, while maintaining highly comparable prediction performance with the two.

The simplicity of the GCN architecture allows for a natural interpretation of graph convolutions in terms of integral transforms. Such a view, yet, generalizes to many graph models whose formulations are based on first-order neighborhoods, examples of which include MoNet that applies to (meshed) manifolds (Monti et al., 2017), as well as many message-passing neural networks (see e.g., Scarselli et al. (2009); Gilmer et al. (2017)). The proposed work elucidates the basic Monte Carlo ingredients for consistently estimating the integrals. When generalizing to other networks aforementioned, an additional effort is to investigate whether and how variance reduction may improve the estimator, a possibly rewarding avenue of future research.

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

## A    PROOFS

*Proof of Theorem 1.*  Because the samples $u_j^{(0)}$ are iid, by the strong law of large numbers,

$$\tilde{h}_{t_1}^{(1)}(v) = \frac{1}{t_0} \sum_{j=1}^{t_0} \hat{A}(v, u_j^{(0)}) h^{(0)}(u_j^{(0)}) W^{(0)}$$

converges almost surely to $\tilde{h}^{(1)}(v)$.  Then, because the activation function $\sigma$ is continuous, the continuous mapping theorem implies that $h_{t_1}^{(1)}(v) = \sigma(\tilde{h}_{t_1}^{(1)}(v))$ converges almost surely to $h^{(1)}(v) = \sigma(\tilde{h}^{(1)}(v))$.  Thus, $\int \hat{A}(v, u) h_{t_1}^{(1)}(u) W^{(1)} \, dP(u)$ converges almost surely to $\tilde{h}^{(2)}(v) = \int \hat{A}(v, u) h^{(1)}(u) W^{(1)} \, dP(u)$, where note that the probability space is with respect to the 0th layer and hence has nothing to do with that of the variable $u$ or $v$ in this statement.  Similarly,

$$\tilde{h}_{t_2}^{(2)}(v) = \frac{1}{t_1} \sum_{j=1}^{t_1} \hat{A}(v, u_j^{(1)}) h_{t_1}^{(1)}(u_j^{(1)}) W^{(1)}$$

converges almost surely to $\int \hat{A}(v, u) h_{t_1}^{(1)}(u) W^{(1)} \, dP(u)$ and thus to $\tilde{h}^{(2)}(v)$.  A simple induction completes the rest of the proof.    □

*Proof of Proposition 2.*  Conditioned on $v$, the expectation of $y(v)$ is

$$\mathrm{E}[y(v)|v] = \int \hat{A}(v, u) x(u) \, dP(u) = e(v), \tag{11}$$

and the variance is $1/t$ times that of $\hat{A}(v, u) x(u)$, i.e.,

$$\mathrm{Var}\{y(v)|v\} = \frac{1}{t} \left( \int \hat{A}(v, u)^2 x(u)^2 \, dP(u) - e(v)^2 \right). \tag{12}$$

Instantiating (11) and (12) with iid samples $v_1, \ldots, v_s \sim P$ and taking variance and expectation in the front, respectively, we obtain

$$\mathrm{Var}\left\{ \mathrm{E}\left[ \frac{1}{s} \sum_{i=1}^{s} y(v_i) \middle| v_1, \ldots, v_s \right] \right\} = \mathrm{Var}\left\{ \frac{1}{s} \sum_{i=1}^{s} e(v_i) \right\} = \frac{1}{s} \int e(v)^2 \, dP(v) - \frac{1}{s} \left( \int e(v) \, dP(v) \right)^2,$$

and

$$\mathrm{E}\left[ \mathrm{Var}\left\{ \frac{1}{s} \sum_{i=1}^{s} y(v_i) \middle| v_1, \ldots, v_s \right\} \right] = \frac{1}{st} \iint \hat{A}(v, u)^2 x(u)^2 \, dP(u) \, dP(v) - \frac{1}{st} \int e(v)^2 \, dP(v).$$

Then, applying the law of total variance

$$\mathrm{Var}\left\{ \frac{1}{s} \sum_{i=1}^{s} y(v_i) \right\} = \mathrm{Var}\left\{ \mathrm{E}\left[ \frac{1}{s} \sum_{i=1}^{s} y(v_i) \middle| v_1, \ldots, v_s \right] \right\} + \mathrm{E}\left[ \mathrm{Var}\left\{ \frac{1}{s} \sum_{i=1}^{s} y(v_i) \middle| v_1, \ldots, v_s \right\} \right],$$

we conclude the proof.    □

*Proof of Theorem 3.* Conditioned on $v$, the variance of $y_Q(v)$ is $1/t$ times that of

$$\hat{A}(v, u)x(u)\frac{dP(u)}{dQ(u)} \quad \text{(where } u \sim Q\text{)},$$

i.e.,

$$\text{Var}\{y_Q(v)|v\} = \frac{1}{t}\left(\int \frac{\hat{A}(v, u)^2 x(u)^2 dP(u)^2}{dQ(u)} - e(v)^2\right).$$

Then, following the proof of Proposition 2, the overall variance is

$$\text{Var}\{G_Q\} = R + \frac{1}{st}\iint \frac{\hat{A}(v, u)^2 x(u)^2 \, dP(u)^2 \, dP(v)}{dQ(u)} = R + \frac{1}{st}\int \frac{b(u)^2 x(u)^2 dP(u)^2}{dQ(u)}.$$

Hence, the optimal $dQ(u)$ must be proportional to $b(u)|x(u)| \, dP(u)$. Because it also must integrate to unity, we have

$$dQ(u) = \frac{b(u)|x(u)| \, dP(u)}{\int b(u)|x(u)| \, dP(u)},$$

in which case

$$\text{Var}\{G_Q\} = R + \frac{1}{st}\left[\int b(u)|x(u)| \, dP(u)\right]^2.$$

$\square$

*Proof of Proposition 4.* Conditioned on $v$, the variance of $y_Q(v)$ is $1/t$ times that of

$$\hat{A}(v, u)x(u)\frac{dP(u)}{dQ(u)} = \frac{\hat{A}(v, u)\,\text{sgn}(x(u))}{b(u)}\int b(u)|x(u)| \, dP(u),$$

i.e.,

$$\text{Var}\{y_Q(v)|v\} = \frac{1}{t}\left(\left[\int b(u)|x(u)| \, dP(u)\right]^2 \int \frac{\hat{A}(v, u)^2}{b(u)^2} \, dQ(u) - e(v)^2\right).$$

The rest of the proof follows that of Proposition 2.

$\square$

## B  ADDITIONAL EXPERIMENT DETAILS

### B.1  BASELINES

**GCN**: The original GCN cannot work on very large graphs (e.g., Reddit). So we modified it into a batched version by simply removing the sampling in our FastGCN (i.e., using all the nodes instead of sampling a few in each batch). For relatively small graphs (Cora and Pubmed), we also compared the results with the original GCN.

**GraphSAGE**: For training time comparison, we use GraphSAGE-GCN that employs GCN as the aggregator. It is also the fastest version among all choices of the aggregators. For accuracy comparison, we also compared with GraphSAGE-mean. We used the codes from `https://github.com/williamleif/GraphSAGE`. Following the setting of Hamilton et al. (2017), we use two layers with neighborhood sample sizes $S_1 = 25$ and $S_2 = 10$. For fair comparison with our method, the batch size is set to be the same as FastGCN, and the hidden dimension is 128.

### B.2  EXPERIMENT SETUP

**Datasets:** The Cora and Pubmed data sets are from `https://github.com/tkipf/gcn`. As we explained in the paper, we kept the validation index and test index unchanged but changed the training index to use all the remaining nodes in the graph. The Reddit data is from `http://snap.stanford.edu/graphsage/`.

**Experiment Setting:** We preformed hyperparameter selection for the learning rate and model dimension. We swept learning rate in the set {0.01, 0.001, 0.0001}. The hidden dimension of Fast-GCN for Reddit is set as 128, and for the other two data sets, it is 16. The batch size is 256

for Cora and Reddit, and 1024 for Pubmed. Dropout rate is set as 0. We use Adam as the optimization method for training. In the test phase, we use the trained parameters and all the graph nodes instead of sampling. For more details please check our codes in a temporary git repository `https://github.com/matenure/FastGCN`.

**Hardware:** Running time is compared on a single machine with 4-core 2.5 GHz Intel Core i7, and 16G RAM.

## C  ADDITIONAL EXPERIMENTS

### C.1  TRAINING TIME COMPARISON

Figure 3 in the main text compares the per-batch training time for different methods. Here, we list the total training time for reference. It is impacted by the convergence of SGD, whose contributing factors include learning rate, batch size, and sample size. See Table 4. Although the orders-of-magnitude speedup of per-batch time is slightly weakened by the convergence speed, one still sees a substantial advantage of the proposed method in the overall training time. Note that even though the original GCN trains faster than the batched version, it does not scale because of memory limitation. Hence, a fair comparison should be gauged with the batched version. We additionally show in Figure 4 the evolution of prediction accuracy as training progresses.

Table 4: Total training time (in seconds).

|                | Cora | Pubmed | Reddit  |
| -------------- | ---- | ------ | ------- |
| FastGCN        | 2.7  | 15.5   | 638.6   |
| GraphSAGE-GCN  | 72.4 | 259.6  | 3318.5  |
| GCN (batched)  | 6.9  | 210.8  | 58346.6 |
| GCN (original) | 1.7  | 21.4   | NA      |

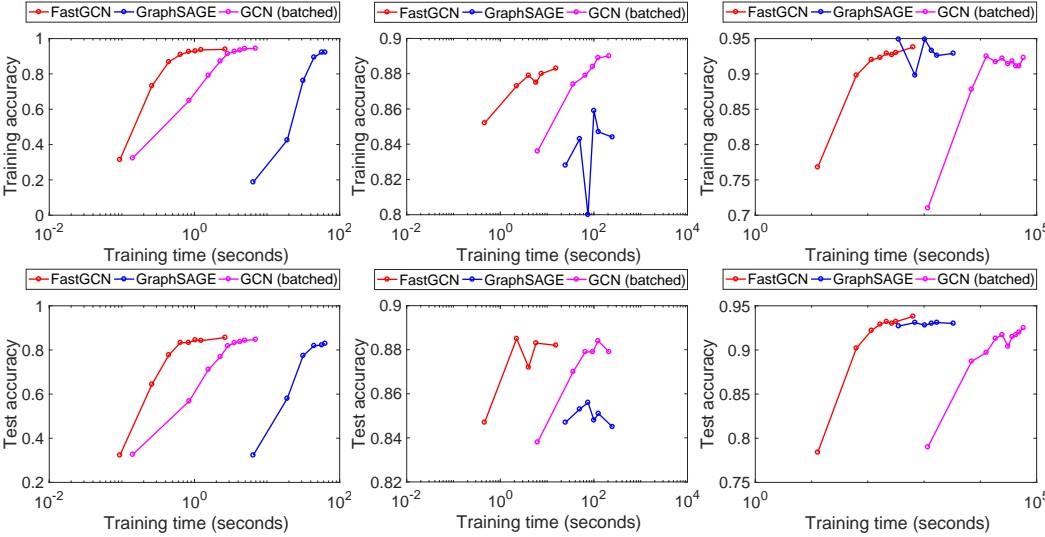

Figure 4: Training/test accuracy versus training time. From left to right, the data sets are Cora, Pubmed, and Reddit, respectively.

### C.2  ORIGINAL DATA SPLIT FOR CORA AND PUBMED

As explained in Section 4, we increased the number of labels used for training in Cora and Pubmed, to align with the supervised learning setting of Reddit. For reference, here we present results by using the original data split with substantially fewer training labels. We also fork a separate version of FastGCN, called FastGCN-transductive, that uses both training and test data for learning. See Table 5.

The results for GCN are consistent with those reported by Kipf & Welling (2016a). Because labeled data are scarce, the training of GCN is quite fast. FastGCN beats it only on Pubmed. The accuracy results of FastGCN are inferior to GCN, also because of the limited number of training labels. The transductive version FastGCN-transductive matches the accuracy of that of GCN. The results for GraphSAGE are curious. We suspect that the model significantly overfits the data, because perfect training accuracy (i.e., 1) is attained.

One may note a subtlety that the training of GCN (original) is slower than what is reported in Table 4, even though fewer labels are used here. The reason is that we adopt the same hyperparameters as in Kipf & Welling (2016a) to reproduce the F1 scores of their work, whereas for Table 4, a better learning rate is found that boosts the performance on the new split of the data, in which case GCN (original) converges faster.

Table 5: Total training time and test accuracy for Cora and Pubmed, original data split. Time is in seconds.

|  | Cora | | Pubmed | |
| --- | --- | --- | --- | --- |
|  | Time | F1 | Time | F1 |
| FastGCN | 2.52 | 0.723 | 0.97 | 0.721 |
| FastGCN-transductive | 5.88 | 0.818 | 8.97 | 0.776 |
| GraphSAGE-GCN | 107.95 | 0.334 | 39.34 | 0.386 |
| GCN (original) | 2.18 | 0.814 | 32.65 | 0.795 |

## D  CONVERGENCE

Strictly speaking, the training algorithms proposed in Section 3 do not precisely follow the existing theory of SGD, because the gradient estimator, though consistent, is biased. In this section, we fill the gap by deriving a convergence result. Similar to the case of standard SGD where the convergence rate depends on the properties of the objective function, here we analyze only a simple case; a comprehensive treatment is out of the scope of the present work. For convenience, we will need a separate system of notations and the same notations appearing in the main text may bear a different meaning here. We abbreviate "with probability one" to "w.p.1" for short.

We use $f(x)$ to denote the objective function and assume that it is differentiable. Differentiability is not a restriction because for the nondifferentiable case, the analysis that follows needs simply change the gradient to the subgradient. The key assumption made on $f$ is that it is $l$-strictly convex; that is, there exists a positive real number $l$ such that

$$f(x) - f(y) \geq \langle \nabla f(y), x - y \rangle + \frac{l}{2} \|x - y\|^2, \tag{13}$$

for all $x$ and $y$. We use $g$ to denote the gradient estimator. Specifically, denote by $g(x; \xi_N)$, with $\xi_N$ being a random variable, a strongly consistent estimator of $\nabla f(x)$; that is,

$$\lim_{N \to \infty} g(x; \xi_N) = \nabla f(x) \quad \text{w.p.1}.$$

Moreover, we consider the SGD update rule

$$x_{k+1} = x_k - \gamma_k \, g(x_k; \xi_N^{(k)}), \tag{14}$$

where $\xi_N^{(k)}$ is an indepedent sample of $\xi_N$ for the $k$th update. The following result states that the update converges on the order of $O(1/k)$.

**Theorem 5.** *Let $x^*$ be the (global) minimum of $f$ and assume that $\|\nabla f(x)\|$ is uniformly bounded by some constant $G > 0$. If $\gamma_k = (lk)^{-1}$, then there exists a sequence $B_k$ with*

$$B_k \leq \frac{\max\{\|x_1 - x^*\|^2, \ G^2/l^2\}}{k}$$

*such that $\|x_k - x^*\|^2 \to B_k$ w.p.1.*

*Proof.* Expanding $\|x_{k+1} - x^*\|^2$ by using the update rule (14), we obtain

$$\|x_{k+1} - x^*\|^2 = \|x_k - x^*\|^2 - 2\gamma_k\langle g_k, x_k - x^*\rangle + \gamma_k^2\|g_k\|^2,$$

where $g_k \equiv g(x_k; \xi_N^{(k)})$. Because for a given $x_k$, $g_k$ converges to $\nabla f(x_k)$ w.p.1, we have that conditioned on $x_k$,

$$\|x_{k+1} - x^*\|^2 \to \|x_k - x^*\|^2 - 2\gamma_k\langle \nabla f(x_k), x_k - x^*\rangle + \gamma_k^2\|\nabla f(x_k)\|^2 \quad \text{w.p.1.} \quad (15)$$

On the other hand, applying the strict convexity (13), by first taking $x = x_k, y = x^*$ and then taking $x = x^*, y = x_k$, we obtain

$$\langle \nabla f(x_k), x_k - x^*\rangle \geq l\|x_k - x^*\|^2. \quad (16)$$

Substituting (16) to (15), we have that conditioned on $x_k$,

$$\|x_{k+1} - x^*\|^2 \to C_k \quad \text{w.p.1}$$

for some

$$C_k \leq (1 - 2l\gamma_k)\|x_k - x^*\|^2 + \gamma_k^2 G^2 = (1 - 2/k)\|x_k - x^*\|^2 + G^2/(l^2 k^2). \quad (17)$$

Now consider the randomness of $x_k$ and apply induction. For the base case $k = 2$, the theorem clearly holds with $B_2 = C_1$. If the theorem holds for $k = T$, let $L = \max\{\|x_1 - x^*\|^2, \ G^2/l^2\}$. Then, taking the probabilistic limit of $x_T$ on both sides of (17), we have that $C_T$ converges w.p.1 to some limit that is less than or equal to $(1 - 2/T)(L/T) + G^2/(l^2 T^2) \leq L/(T + 1)$. Letting this limit be $B_{T+1}$, we complete the induction proof. □

