# OpenReview forum: "FastGCN: Fast Learning with Graph Convolutional Networks via Importance Sampling"
_ICLR.cc/2018/Conference — Accept (Poster)_

### Official Review · AnonReviewer3 · 2017-11-23
**present a novel view of GCN that leads to scalable GCN further with importance sampling for variance reduction**

**Rating:** 7
**Confidence:** 2

**Review:**

The paper presents a novel view of GCN that interprets graph convolutions as integral transforms of embedding functions. This addresses the issue of lack of sample independence in training and allows for the use of Monte Carlo methods. It further explores variance reduction to speed up training via importance sampling.  The idea comes with theoretical support and experimental studies.

Some questions are as follows:

1) could you elaborate on n/t_l  in (5) that accounts for the normalization difference between matrix form (1) and the integral form (2) ?

2) In Prop.2., there seems no essential difference between the two parts, as e(v) also depends on how the u_j's are sampled.

3) what loss g is used in experiments?

---

> ### Author Response · Authors · 2017-12-18
> **response, discussion, revision**
>
> Thank you very much for the questions. Please find our responses in the following. We hope that your confusions are now cleared.
>
> >>> could you elaborate on n/t_l  in (5) that accounts for the normalization difference between matrix form (1) and the integral form (2) ?
>
> For (2), a probability measure must integrate to unity. On other hand, for the matrix form (1), the matrix products will explode when the matrix size becomes larger and larger. What is lacking is a factor of n that normalizes (1).
>
> In fact, such an issue could be more principledly explained in the context of importance sampling in the subsection that follows. Note the displayed formula in Algorithm 2. Without using importance sampling, the denominator q(u_j^{(l)}) is simply 1/n, hence simplified to Algorithm 1.
>
> >>> In Prop.2., there seems no essential difference between the two parts, as e(v) also depends on how the u_j's are sampled.
>
> It is true that e(v) is an integral in the u space. What we meant on the other hand is that if we change the way the u_j’s are sampled, the variance of G will respectively change. The specific amount of change (compare Proposition 2, Theorem 3, and Proposition 4) happens to the second term, leaving the first term R untouched. Please see the derivation (proof) in the appendix.
>
> >>> what loss g is used in experiments?
>
> Following GCN and GraphSAGE, the loss is the cross entropy.

---

### Official Review · AnonReviewer1 · 2017-11-27
**Solid idea, excellent presentation, questions about experiments**

**Rating:** 7
**Confidence:** 4

**Review:**

The paper focuses on the recently graph convolutional network (GCN) framework.
They authors identify a couple of issues with GCN: the fact that both training and test data need to be present at training time, making it transductive in nature and the fact that the notion of ‘neighborhood’ grows as the signal propagates through the network. The latter implies that GCNs can have a large memory footprint, making them impractical in certain cases.
The authors propose an alternative formulation that interprets the signals as vertex embedding functions; it also interprets  graph convolutions as integral transforms of said functions.
Starting from mini-batches consisting purely of training data (during training) each layer performs Monte Carlo sampling on the vertices to approximate the embedding functions.
They show that this estimator is consistent and can be used for training the proposed architecture, FastGCN, via standard SGD.
Finally, they analyze the estimator’s variance and propose an importance-sampling based estimator that has minimal layer-to-layer variance.
The experiments demonstrate that FastGCN is much faster than the alternatives, while suffering a small accuracy penalty.

This is a very good paper. The ideas are solid, the writing is excellent and the results convincing. I have a few comments and concerns listed below.

Comments:
1. I agree with the anonymous commenter that the authors should provide detailed description of their experimental setup.
2. The timing of GraphSAGE on Cora is bizarre. I’m even slightly suspicious that something might have been amiss in your setup. It is by far the smallest dataset. How do you explain GraphSAGE performing so much worse on Cora than on the bigger Pubmed and Reddit datasets? It is also on Cora that GraphSAGE seems to yield subpar accuracy, while it wins the other two datasets.
3. As a concrete step towards grounding the proposed method on state of the art results, I would love to see at least one experiment with the same (original) data splits used in previous papers. I understand that semi-supervised learning is not the purpose of this paper, however matching previous results would dispel any concerns about setup/hyperparameter mismatch.
4. Another thing missing is an exploration (or at least careful discussion) as to why FastGCN performs worse than the other methods in terms of accuracy and how much that relative penalty can be.

Minor comments:
5. Please add label axes to Figure 2; currently it is very hard to read. Also please label the y axis in Figure 3.
6. The notation change in Section 3.1 was well intended, however I feel like it slowed me down significantly while reading the paper. I had already absorbed the original notation and had to go back and forth to translate to the new one.

---

> ### Author Response · Authors · 2017-12-18
> **response, discussion, revision**
>
> Thank you very much for your positive comments. Please find our responses and summary of revisions in the following. Your reviews are cited with >>>.
>
> >>> I agree with the anonymous commenter that the authors should provide detailed description of their experimental setup.
>
> We have inserted details regarding the train/val/test split concerned by the anonymous commenter, in the main text. Additional experiments were included in the appendix.
>
> >>> The timing of GraphSAGE on Cora is bizarre. I’m even slightly suspicious that something might have been amiss in your setup. It is by far the smallest dataset. How do you explain GraphSAGE performing so much worse on Cora than on the bigger Pubmed and Reddit datasets? It is also on Cora that GraphSAGE seems to yield subpar accuracy, while it wins the other two datasets.
>
> We double checked the code and reran the experiments but did not spot abnormality. We encourage the reviewer to checkout our code from the anonymous github and verify. Here are our thoughts: For training time, GraphSAGE uses sampling so the time is independent of the graph size. The times across data sets should be comparable since sample sizes are comparable. Fluctuations are normal. For accuracy, we did another round of hyperparameter tuning and found that the F1 score on Cora can be improved. The newer results were updated to the table in Figure 3. However, these better results are still subpar compared with those of GCN and FastGCN.
>
> >>> As a concrete step towards grounding the proposed method on state of the art results, I would love to see at least one experiment with the same (original) data splits used in previous papers. I understand that semi-supervised learning is not the purpose of this paper, however matching previous results would dispel any concerns about setup/hyperparameter mismatch.
>
> We have included an additional experiment in the appendix; see Section C.2. The results for GCN are consistent with those reported by Kipf and Welling. We have not seen reported results for GraphSAGE on these data sets; our results suggest way inferior performance. It is suspected that the model significantly overfits the data, because training accuracy is 1. For the proposed FastGCN, it also performs inferior to GCN, probably because of the very limited number of training labels. We fork a different version, called FastGCN-transductive, which uses both training and test data for learning (hence falling back to the transductive setting of GCN). The results of FastGCN-transductive match those of GCN.
>
> >>> Another thing missing is an exploration (or at least careful discussion) as to why FastGCN performs worse than the other methods in terms of accuracy and how much that relative penalty can be.
>
> We would argue that the accuracy results of FastGCN are quite comparable with the best of other methods. The loss of accuracy is even smaller than the difference among the several aggregators proposed for GraphSAGE. The improvement in running time outweighs such a minimal loss.
>
> >>> Minor comments:
> >>> Please add label axes to Figure 2; currently it is very hard to read. Also please label the y axis in Figure 3.
>
> Done.
>
> >>> The notation change in Section 3.1 was well intended, however I feel like it slowed me down significantly while reading the paper. I had already absorbed the original notation and had to go back and forth to translate to the new one.
>
> It is an unfortunate compromise, because the notations developed so far have become too cumbersome. If we carry the subscripts and superscripts to the rest of the paper, the digestion of the math is possibly even harder.

---

> ### Author Response · Authors · 2017-12-20
> **additional update**
>
> We would like to update that the authors of GraphSAGE offered an improved implementation of their codes for small graphs. Now the timing of GraphSAGE on Cora is more favorable. Please see the last paragraph of Section 4 and also Table 3.

---

### Official Review · AnonReviewer2 · 2017-11-27
**Interesting ideas, but I have both theoretical and practical concerns**

**Rating:** 6
**Confidence:** 4

**Review:**

Update:

I have read the rebuttal and the revised manuscript. Additionally I had a brief discussion with the authors regarding some aspects of their probabilistic framework. I think that batch training of GCN is an important problem and authors have proposed an interesting solution to this problem. I appreciated all the work authors put into the revision. In this regard, I have updated my rating. However, I am not satisfied with how the probabilistic problem formulation was presented in the paper. I would appreciate if authors were more upfront about the challenges of the problem they formulated and limitations of their results. I briefly summarize the key missing points below, although I acknowledge that solution to such questions is out of scope of this work.

1. Sampling of graph nodes from P is not iid. Every subsequent node can not be equal to any of the previous nodes. Hence, the distribution changes and subsequent nodes are dependent on previous ones. However, exchangeability could be a reasonable assumption to make as order (in the joint distribution) does not matter for simple choices of P. Example: let V be {1,2,3} and P a uniform distribution. First node can be any of the {1,2,3}, second node given first (suppose first node is '2') is restricted to {1,3}. There is clearly a dependency and change of distribution.

2. Theorem 1 is proven under the assumption that it is possible to sample from P and utilize Monte Carlo type argument. However, in practice, sampling is done from a uniform distribution over observed samples. Also, authors suggest that V may be infinite. Recall that for Monte Carlo type approaches to work, sampling distribution is ought to contain support of the true distribution. Observed samples (even as sample size goes to infinity) will never be able to cover an infinite V. Hence, Theorem 1 will never be applicable (for the purposes of evaluating population loss). Also note that this is different from a more classical case of continuous distributions, where sampling from a Gaussian, for instance, will cover any domain of true distribution. In the probabilistic framework defined by the authors it is impossible to cover domain of P, unless whole V is observed.

----------------------------------------------------------------------
This work addresses a major shortcoming of recently popularized GCN. That is, when the data is equipped with the graph structure, classic SGD based methods are not  straightforward to apply. Hence it is not clear how to deal with large datasets (e.g., Reddit). Proposed approach uses an adjacency based importance sampling distribution to select only a subset of nodes on each GCN layer. Resulting loss estimate is shown to be consistent and its gradient is used to perform the weight updates.

Proposed approach is interesting and the direction of the work is important given recent popularity of the GCN. Nonetheless I have two major question and would be happy to revisit my score if at least one is addressed.

Theory:
SGD requires an unbiased estimate of the gradient to converge to the global optima in the convex loss case. Here, the loss estimate is shown to be consistent, but not guaranteed to be unbiased and nothing is said about the gradient in Algorithm 1. Could you please provide some intuition about the gradient estimate? I might not be familiar with some relevant results, but it appears to me that Algorithm 1 will not converge to the same solution as full data GD would.

Practice:
Per batch timings in Fig. 3 are not enough to argue that the method is faster as it might have poor convergence properties overall. Could you please show the train/test accuracies against training time for all compared methods?

Some other concerns and questions:
- It is not quite cleat what P is. You defined it as distribution over vertices of some (potentially infinite) population graph. Later on, sampling from P becomes equivalent to uniform sampling over the observed nodes. I don't see how you can define P over anything outside of the training nodes (without defining loss on the unobserved data), as then you would be sampling from a distribution with 0 mass on the parts of the support of P, and this would break the Monte Carlo assumptions.
- Weights disappeared in the majority of the analysis. Could you please make the representation more consistent.
- a(v,u) in Eq. 2 and A(v,u) in Eq. 5 are not defined. Do they both correspond to entries of the (normalized) adjacency?

---

> ### Author Response · Authors · 2017-12-18
> **response, discussion, revision**
>
> We appreciate very much your critical comments. Please find our responses and summary of revisions in the following. Your reviews are cited with >>>. We hope that the edited version may clear the confusion and you enjoy the paper as other reviewers do :)
>
> >>> Theory:
> >>> SGD requires an unbiased estimate of the gradient to converge to the global optima in the convex loss case. Here, the loss estimate is shown to be consistent, but not guaranteed to be unbiased and nothing is said about the gradient in Algorithm 1. Could you please provide some intuition about the gradient estimate? I might not be familiar with some relevant results, but it appears to me that Algorithm 1 will not converge to the same solution as full data GD would.
>
> The consistency of the gradient estimator simply follows that of the loss estimator, if the differential operator is continuous. Hence, the essential question is whether SGD converges if the gradient estimator is consistent but not unbiased. We have developed a convergence theory in the appendix (see Section D) for our algorithms. Generally speaking, the convergence rate is the same as the case of unbiased gradient estimator.
>
> >>> Practice:
> >>> Per batch timings in Fig. 3 are not enough to argue that the method is faster as it might have poor convergence properties overall. Could you please show the train/test accuracies against training time for all compared methods?
>
> We found that the convergence speed between GCN and FastGCN was empirically similar, whereas GraphSAGE appears to converge much faster. Coupled with the per-epoch cost, overall FastGCN still wins with a substantial margin. We have inserted a section in the appendix to cover the total training time as well as the accuracy. Please see Section C.1 and particularly Table 3 and Figure 4.
>
> >>> Some other concerns and questions:
> >>> It is not quite clear what P is. You defined it as distribution over vertices of some (potentially infinite) population graph. Later on, sampling from P becomes equivalent to uniform sampling over the observed nodes. I don't see how you can define P over anything outside of the training nodes (without defining loss on the unobserved data), as then you would be sampling from a distribution with 0 mass on the parts of the support of P, and this would break the Monte Carlo assumptions.
>
> This would be a very interesting excursion. In a sampling framework that we are settling with (all being traced back to what empirical risk minimization means for graphs), P is an abstract probability measure for the graph nodes. For the sake of simplicity imagine an infinite graph (just like the usual vectorial case where the input space is d-dimensional Euclidean). Some graph nodes are sampled for training and some others are used for validation and testing. P is the underlying (unknown) probability distribution that one uses for sampling.
>
> The uniform sampling mentioned later is a separate story. Suppose that you already have a sample (i.e., the training set). Note that “a sample” here means a collection of data points drawn iid from a population. And you want to estimate some properties of the population (i.e., the expected loss). Bootstrapping is a scheme that subsamples the given sample for performing inference on the unknown population. This corresponds to using a mini-batch of the training set to estimate the expected loss. The most straightforward approach for bootstrapping is a uniform subsampling with or without replacement. Importance (sub)sampling as we use later may yield a better estimate.
>
> >>> Weights disappeared in the majority of the analysis. Could you please make the representation more consistent.
>
> We reexamined the whole paper and included the weights as appropriate. Since they are linear, the overall theory and conclusions remain valid.
>
> >>> a(v,u) in Eq. 2 and A(v,u) in Eq. 5 are not defined. Do they both correspond to entries of the (normalized) adjacency?
>
> Yes they do. Text was edited.

---

> > ### Comment · AnonReviewer2 · 2018-01-05
> > **Regarding P and bootstrapping comparison**
> >
> > Sampling of train and test vertices does not appear iid to me. Collection of graph vertices in the test set is guaranteed to be disjoint from the train set, which introduces the dependency. Bootstrapping, on the other hand, requires independence to produce meaningful estimates and I don not see it being applicable when support of train and test data is predefined to be disjoint. I would appreciate a more rigorous explanation.

---

> > > ### Author Response · Authors · 2018-01-05
> > > **RE: Regarding P and bootstrapping comparison**
> > >
> > > The confusion may be cleared by considering the difference between transductive learning and inductive learning. The former is the setting on which the original GCN is based, whereas our work extends to the latter. In the transductive setting, the training set and the test set, often disjoint, are used for learning. However, in the inductive setting, only the training set is used. Hence, the sampling that yields the training set, as well as the bootstrapping from the training set, has nothing to do with the test set. The iid assumption poses no contradiction, to our view.

---

> > > > ### Comment · AnonReviewer2 · 2018-01-05
> > > > **RE: RE: Regarding P and bootstrapping comparison**
> > > >
> > > > That is why I initially mentioned that you need to somehow define the loss on the unlabeled examples. There is a single set of nodes that was generated from P (which is training and test data together), however some of the nodes are unlabeled. When you are doing the bootstrapping, the unlabeled nodes should also be considered during the sub-sampling (and have a non-zero probability to be selected).

---

> > > > > ### Author Response · Authors · 2018-01-05
> > > > > **RE: RE: RE: Regarding P and bootstrapping comparison**
> > > > >
> > > > > It could be either that we miss your point of “unlabeled data” or you misunderstood our replies. May we solicit a few possible meanings of “unlabeled data” in order that the discussion be more fruitful? Assume that training set and test set are disjoint.
> > > > >
> > > > > 1. “Unlabeled data” means vertices outside the training and test sets. We do need to assume that all vertices in the (possibly infinite) population graph have labels. The distribution P is defined on all vertices.
> > > > >
> > > > > 2. “Unlabeled data” means the test set. Because learning has nothing to do with the test set, the empirical risk should not contain unlabeled data. The training set is obtained from iid sampling and nothing is said about the test set (standard learning theory). Bootstrapping should be done on the training set only. This is the point we made in the past reply.
> > > > >
> > > > > 3. “Unlabeled data” means that some vertices in the training set are unlabeled. Since we are not doing transductive learning, such a situation is precluded.

---

> > ### Comment · AnonReviewer2 · 2018-01-14
> > **Response to rebuttal**
> >
> > I have read the rebuttal and the updated manuscript. I still have several questions left before I revise my review and rating.
> >
> > 1. I agree with statement in Theorem 1, however I am under the impression that you imply that Eq. 4 is a consistent estimator of Eq. 3. This is not true and this is why I was asking about P. Moreover, this does not need to be true for Algorithm 1 to make sense. You want to obtain consistent estimate of the gradient, which is based on the sample loss, not a population loss. And I agree that Eq. 4 is consistent estimate of the sample loss based on your theory, however all preceding to Eq. 4 discussion (Eq. 3 and Theorem 1 in particular) are for the population case and this makes the interpretation of Eq. 4 confusing. Could you please elaborate on what Eq. 4 is estimating.
> >
> > Minor suggestion and comment:
> > 1. You added the plot for test accuracy convergence in the Appendix, which is helpful. Although convergence of an optimization algorithm is better evaluated on the loss function it is optimizing - could you please add train accuracy convergence as well.
> > 2. Please use either a(u,v) or A(u,v) for entries of the adjacency (Eq. 2 and Eq. 5)

---

> > > ### Author Response · Authors · 2018-01-16
> > > **paper updated**
> > >
> > > We appreciate your patience. For the minor comments, we have included the training accuracy plots in Fig 4 (appendix) and changed the lower case a to upper case.
> > >
> > > Regarding Eq 4, it is true that in practice the algorithm needs to take care of only the sample loss. We also agree that if a sample is considered fixed, then bootstrapping cannot go beyond the precision of whatever quantity the sample estimates. We do want to point out, on the other hand, that if the sample size is considered varying to infinity and likewise for the resample size, then the consistent property of the estimator Eq 4 can be established. An intuitive explanation is something like-- if Eq 4 converges to the sample loss with probability one and the sample loss converges to the population loss with probability one, then Eq 4 converges to the population loss with probability one. Of course for the rigorous argument we need to take two dimensional limits.

---

### Official Review · AnonReviewer4 · 2017-12-01
**Fast solution for the memory bottleneck issue in graph neural networks**

**Rating:** 8
**Confidence:** 4

**Review:**

This paper addresses the memory bottleneck problem in graph neural networks and proposes a novel importance sampling scheme that is based on sampling vertices (instead of sampling local neighbors as in [1]). Experimental results demonstrate a significant speedup in per-batch training time compared to previous works while retaining similar classification accuracy on standard benchmark datasets.

The paper is well-written and proposes a simple, elegant, and well-motivated solution for the memory bottleneck issue in graph neural networks.

I think that this paper mostly looks solid, but I am a bit worried about the following assumption: “Specifically, we interpret that graph vertices are iid samples of some probability distribution”. As graph vertices are inter-connected and inter-dependent across edges of the graph, this iid assumption might be too strong. A short comment on why the authors take this particular interpretation would be helpful.

In the abstract the authors write: “Such a model [GCN], however, is transductive in nature because parameters are learned through convolutions with both training and test data.” — as demonstrated in Hamilton et al. (2017) [1], this class of models admits inductive learning as well as transductive learning, so the above statement is not quite accurate.

Furthermore, a comment on whether this scheme would be useful for alternative graph neural network architectures, such as the one in MoNet [2] or the generic formulation of the original graph neural net [3] (nicely summarized in Gilmer et al. (2017) [4]) would be insightful (and would make the paper even stronger).

I am very happy to see that the authors provide the code together with the submission (using an anonymous GitHub repository). The authors mention that “The code of GraphSAGE is downloaded from the accompany [sic] website, whereas GCN is self implemented.“ - Looking at the code it looks to me, however, as if it was based on the implementation by the authors of [5].

The experimental comparison in terms of per-batch training time looks very impressive, yet it would be good to also include a comparison in terms of total training time per model (e.g. in the appendix). I quickly checked the provided implementation for FastGCN on Pubmed and compared it against the GCN implementation from [5], and it looks like the original GCN model is roughly 30% faster on my laptop (no batched training). This is not very surprising, as a fair comparison should involve batched training for both approaches. Nonetheless it would be good to include these results in the paper to avoid confusion.

Minor issues:
- The notation of the limit in Theorem 1 is a bit unclear. I assume the limit is taken to infinity with respect to the number of samples.
- There are a number of typos throughout the paper (like “oppose to” instead of “opposed to”), these should be fixed in the revision.
- It would be better to summarize Figure 3 (left) in a table, as the smaller values are difficult to read off the chart.

Overall, I think that this paper can be accepted. The proposed scheme is a simple drop-in replacement for the way adjacency matrices are prepared in current implementations of graph neural nets and it promises to solve the memory issue of previous works while being substantially faster than the model in [1]. I expect the proposed approach to be useful for most graph neural network models.

UPDATE: I would like to thank the authors for their detailed response and for adding additional experimental evaluation. My initial concerns have been addressed and I can fully recommend acceptance of this paper.

[1] W.L. Hamilton, R. Ying, J. Leskovec, Inductive Representation Learning on Large Graphs, NIPS 2017
[2] F. Monti, D. Boscaini, J. Masci, E. Rodala, J. Svoboda, M.M. Bronstein, Geometric deep learning on graphs and manifolds using mixture model CNNs, CVPR 2017
[3] F. Scarselli, M. Gori, A.C. Tsoi, M. Hagenbuchner, G. Monfardini, The Graph Neural Network Model, IEEE Transactions on Neural Networks, 2009
[4] J. Gilmer, S.S. Schoenholz, P.F. Riley, O. Vinyals, G.E. Dahl, Neural Message Passing for Quantum Chemistry, ICML 2017
[5] T.N. Kipf, M. Welling, Semi-Supervised Classification with Graph Convolutional Networks, ICLR 2017

---

> ### Author Response · Authors · 2017-12-18
> **response, discussion, revision**
>
> Thank you very much for your encouraging comments. Please find our responses and summary of revisions in the following. Your reviews are cited with >>>.
>
> >>> I think that this paper mostly looks solid, but I am a bit worried about the following assumption: “Specifically, we interpret that graph vertices are iid samples of some probability distribution”. As graph vertices are inter-connected and inter-dependent across edges of the graph, this iid assumption might be too strong. A short comment on why the authors take this particular interpretation would be helpful.
>
> The iid assumption was made to be conformant with the standard learning setting that minimizes the empirical risk of iid samples. The motivation was developed at the beginning of Section 3.
>
> >>> In the abstract the authors write: “Such a model [GCN], however, is transductive in nature because parameters are learned through convolutions with both training and test data.” — as demonstrated in Hamilton et al. (2017) [1], this class of models admits inductive learning as well as transductive learning, so the above statement is not quite accurate.
>
> Yes, Hamilton et al. established an extension of GCN to the task of inductive unsupervised learning. For preciseness, we edited our statement. Now it reads: “This model, however, was originally designed to be learned with the presence of both training and test data.”
>
> >>> Furthermore, a comment on whether this scheme would be useful for alternative graph neural network architectures, such as the one in MoNet [2] or the generic formulation of the original graph neural net [3] (nicely summarized in Gilmer et al. (2017) [4]) would be insightful (and would make the paper even stronger).
>
> Thank you very much for suggesting generalize our work to other architectures. Indeed, the simple yet powerful idea of sampling is often applicable to models that are based on first-order neighborhoods. We extended a paragraph in the concluding section to stress this point and also suggested an avenue of future work.
>
> >>> I am very happy to see that the authors provide the code together with the submission (using an anonymous GitHub repository). The authors mention that “The code of GraphSAGE is downloaded from the accompany [sic] website, whereas GCN is self implemented.“ - Looking at the code it looks to me, however, as if it was based on the implementation by the authors of [5].
>
> Yes, the codes of FastGCN are based on the implementation of [5]. We meant that we used the codes of GraphSAGE without change, but implemented our own algorithm and changed the GCN codes to adapt to our problem setting. We have modified the text to clarify the confusion.
>
> >>> The experimental comparison in terms of per-batch training time looks very impressive, yet it would be good to also include a comparison in terms of total training time per model (e.g. in the appendix). I quickly checked the provided implementation for FastGCN on Pubmed and compared it against the GCN implementation from [5], and it looks like the original GCN model is roughly 30% faster on my laptop (no batched training). This is not very surprising, as a fair comparison should involve batched training for both approaches. Nonetheless it would be good to include these results in the paper to avoid confusion.
>
> We have included additional results regarding the total training time in the appendix. Please see Section C.1. Note that for faster convergence, the learning rate of FastGCN has been changed to 0.01 in our codes, so now it is faster than the original GCN model on Pubmed. The accuracy of FastGCN remains the same.
>
> >>> Minor issues:
> >>> The notation of the limit in Theorem 1 is a bit unclear. I assume the limit is taken to infinity with respect to the number of samples.
>
> Yes. Corrected.
>
> >>> There are a number of typos throughout the paper (like “oppose to” instead of “opposed to”), these should be fixed in the revision.
>
> Fixed.
>
> >>> It would be better to summarize Figure 3 (left) in a table, as the smaller values are difficult to read off the chart.
>
> We have increased the font size to make the numbers legible. Also note that the vertical axis is modified to the log10 scale so that orders-of-magnitude improvement can be easily seen. We feel that a bar chart here may be more informative than a table.

---

### Public Comment · (anonymous) · 2017-11-05
**Please be more detailed with your experiment set up**

Exactly how did you change the train/val/test split of the data sets? The accuracy values of GCN reported for Cora and Pubmed are much higher than in all previous work. Why did you not use one of the standard evaluation set ups? (Either the Planetoid split or 10/20 randomly sampled splits)

---

> ### Author Response · Authors · 2017-11-06
> **RE: experiment set up**
>
> Thank you very much for the query of the details. A small summary of the train/val/test split is in the following:
>
> Cora: 2708 nodes in total. Original split 140/500/1000 -> we use 1208/500/1000.
> Pubmed: 19717 nodes in total. Original split 60/500/1000 -> we use 18217/500/1000.
>
> That is, the validation size and test size are unchanged, but we use all the rest data for training, instead of using only a small portion. More specifically, we used the same graph structure and the same input features. Then, we kept the test index unchanged, and selected 500 nodes for validation. All the remaining nodes were used for training.
>
> GCN was originally proposed as a semi-supervised (transductive) method. Hence, only a small portion of the nodes have their labels used for training. Our work, on the other hand, leans toward the supervised (inductive) setting. The main purpose is to demonstrate the scalability and speed of our method. If the training set had only a small number of nodes, the original GCN already works very well and it is not necessary to use our method. Hence, we enlarge the training set by using all available nodes (excluding validation and testing). Moreover, such a split is more coherent with that of the other data set, Reddit, used in another compared work, GraphSAGE.
>
> Because more labels are used for training, it makes sense that the prediction results are better than those in the previous works.
>
> We will edit the paper when allowed to address this question.

---

### Public Comment · (anonymous) · 2017-12-08
**Inductive learning on graph data**

Thank you for the nice work on graph convolutional networks. I am a bit confused on what exactly is "inductive learning on graph data".

To my limited view, the inductive setting is something like: We have a graph G1 for training and another graph G2 for testing. G1 and G2 are separate graphs -- there is no edge connecting the two graphs. We would train a model on G1, and then apply it to predict on G2.

However, by reading the second last paragraph of page 1 and Sec 3.2, it seems like the inductive setting used in this work is different: The training graph G1 might connect to the test graph G2 via some edges. In this case, we would probably propagate the learned embedding on G1 nodes to the nodes in G2 through edges. Isn't it transductive?

Nevertheless, can we extend the proposed FastFCN to the first setting? Do we need to have some "edge sampling" strategy for mini-batch SGD, apart from the "node sampling" strategy proposed in the paper? Thank you!

---

> ### Author Response · Authors · 2017-12-18
> **inductive**
>
> There are slightly different accounts of the distinction between inductive and transductive learning, but it should be well agreed that inductive learning builds a model from the knowledge of labeled data only, and transductive learning from both labeled and unlabeled data.
>
> The transductive setting is highly related to the semi-supervised setting, where only a small portion of the data are known with labels, and hence one may as well incorporate the information of unlabeled data to build a more accurate model. For graphs, it is often the case that the unlabeled vertices happen to be the test data whose labels are awaiting for prediction. Of course, such an understanding is based on the assumption that the given graph is fixed and not evolving (at least no new vertices are added in).
>
> In our work, we find that it would be easier to build a consistent theory and draw connections with risk minimization (which is the standard learning theory), if we think about the given graph as a piece of a larger, possibly infinite, graph. In this vein, observed vertices are given labels and there are unobserved ones whose labels we want to predict later on. In other words, the proposed work generalizes GCN to the supervised and inductive setting, where unlabeled vertices are not used for training.
>
> So, to answer your question, what we are proposing indeed fits your first setting, because the edges between the labeled vertices and the unlabeled ones never enter training (but they do return for inference).

---

> > ### Public Comment · (anonymous) · 2017-12-19
> > **inductive**
> >
> > Thank you for the feedback. Highly appreciated. I have some follow up questions:
> >
> > 1. As you said, the proposed approach fits my first setting, where we have a graph G1 for training and another graph G2 for testing. G1 and G2 are separate graphs -- there won't be any edge between G1 and G2. How comes "they do return for inference"?
> >
> > 2. As stated in Sec 3.2, "for inference, the embedding of a new vertex may be computed by using the full GCN architecture (1)". It is not clear to me. What is \hat{A} in Eq (1) for inference purpose? Shall I include both training and test vertices in \hat{A}, or shall I just use the test vertices to construct \hat{A}? What if there are edges between training graph G1 and test graph G2?
> >
> > Thank you!

---

> > > ### Author Response · Authors · 2017-12-20
> > > **inductive**
> > >
> > > OK we may have misunderstood your focus of the distinction between the two settings. Overall we would prefer to think of a graph with two (or more) parts, although your view of having two separate graphs and view them in a unified angle is also ok. Yes, the two parts (or the two graphs in your terms) may (or may not) be connected. It does not quite matter whether they are or not. One does not need to think along the lines of embedding propagation. In test time, the input features and the learned weights are the most important things for obtaining the final embedding. If the test part is disconnected from the training part, it merely means that the embeddings of the trained nodes do not affect those of the test nodes. The only connection between training and testing is the weights. The \hat{A} contains both the training part and the test part.

---

### Public Comment · ~William_L._Hamilton1 · 2017-12-19
**Regarding the GraphSAGE comparison**

Full disclosure: I am a lead author of GraphSAGE.

This paper provides some interesting contributions, and it is well-written. However, I do want to raise some points regarding the timing comparison with GraphSAGE, which is quite unfair to GraphSAGE (for reasons I’ll elaborate on below). I’ve also provided an alternative implementation of GraphSAGE that behaves more sensibly on the tiny Cora graph (see below). The unfairness stems from two sources:

1) GraphSAGE is designed for massive graphs (>100,000 nodes), and the public implementation assumes that node neighborhoods in a particular batch don’t overlap too much.

2) The authors use the default sample-size hyperparameters for GraphSAGE and apply the public implementation on very small graphs that it was not designed for.

I’ll focus on the Cora dataset, as this is where the issue is most extreme. The Cora dataset has 2708 nodes, and the authors use GraphSAGE sample-size hyperparameters of S_1=25 and S_2=10, meaning that 10 neighbors are sampled in “layer-1” and 25 in “layer-2”, resulting in 250 sampled neighbors per node. Combined with a batch size of 256, this means that each batch samples 64,000 nodes, which is 23X larger than the entire Cora graph. Moreover, the public implementation of GraphSAGE assumes that the graph is large—we actually designed GraphSAGE with an industry collaboration in mind, with 1+ billion nodes—and most importantly, GraphSAGE assumes that the sampled node neighborhoods do not overlap. As a consequence, the public GraphSAGE code does not take into account repeated neighbors that are sampled within a particular batch (i.e., it assumes the sampled neighborhoods of all nodes are disjoint). On Cora, this means that we are doing ~23X more computation than necessary in this setup, since we have essentially sampled the entire graph 23 times in each batch.

As a high-level point, I don’t recommend using GraphSAGE on such small graphs; it was intended for use in large-graph settings where subsampling is actually necessary. (The entire Cora dataset easily fits in main memory on a chromebook…). However, in cases where one would apply GraphSAGE to such small graphs, the right thing to do would be to modify the code so that it does not do repeat calculation when there is significant overlap between node neighborhoods; this is just an implementation detail, and does not fundamentally change the GraphSAGE algorithm. (But I will reiterate this is still just unnecessary overhead—why subsample neighborhoods when each batch is just going to contain the whole graph anyways?)

Of course, I take responsibility for our public implementation not supporting such small graphs, and for not making it clear that GraphSAGE will essentially break when the input graph is as small as Cora. But for reference, I modified some of my private, experimental pytorch code to handle the Cora case: https://github.com/williamleif/graphsage-simple

Forgive the messiness of the code—it was extracted from a larger private repo and coded in a rush while I am traveling. Running this code on my Macbook Pro (2.9 GHz, Intel Core i5, 16Gb RAM), I get about ~0.05 seconds per batch and a validation F1 of around ~0.85 (using split sizes that are the same as FastGCN). These results are much more sensible and comparable to (Fast)GCN. It is likely still a bit slower than the batched GCN code due to the unnecessary overhead of sampling. It is also possible that there is an error in my code as well, since I am traveling and coded it in a rush, but the results look pretty sensible to me. (The timing difference on Pubmed is much less drastic and in the ballpark of the numbers in the paper.)

To summarize, I think this is an interesting paper and commend the authors on their work---I especially liked the detailed discussion of variance reduction—but I would appreciate if the authors would update their timing comparison with GraphSAGE. I would also appreciate a note that GraphSAGE is not designed for such small graphs and that neighborhood subsampling leads to unnecessary overhead when graphs are that small. Again, I take responsibility for not making this limitation of the public GraphSAGE implementation more clear.

Best regards,
Will

---

> ### Public Comment · ~William_L._Hamilton1 · 2017-12-19
> **A couple other minor points**
>
> A couple other minor points:
>
> - On such small graphs, it does not make sense to sample at test/inference time. Again, the old GraphSAGE repo does not easily support this, but the code I linked above does. This is likely the cause of GraphSAGE-GCN doing slightly worse in terms of F1.
>
> - The idea of sampling vertices at each layer makes sense in small graphs or dense graphs. However, I think it might become problematic in massive, sparse graphs. For example, in industry applications with 1+ billion nodes (or sparse graphs with many disconnected components), I would think the odds of a node in a batch having a neighbor in the sampled set will get quite small, unless the sampled set grows proportionally large in size. (Please correct me if I am missing something here).
>
> Again, I really appreciate the authors' hard work, and the quality of their presentation.
>
> Cheers,
> Will

---

> > ### Author Response · Authors · 2017-12-20
> > **timings updated**
> >
> > Thanks for clarifying the timing issue of GraphSAGE and providing a new implementation. We appreciate that! We have included the results of the new codes in the paper and made clear that GraphSAGE is not designed for small graphs (see toward the end of Section 4).
> >
> > We do not have access to billion-node graphs that are very sparse and we would certainly love to try on them. Even though the graph is sparse, it might have a powerlaw structure that contributes dense connections crucial for downstream applications. It does not seem easy to expect what would happen in that case and we’ll keep an open eye on it.

---

> > > ### Public Comment · ~William_L._Hamilton1 · 2017-12-20
> > > **thanks for the updates!**
> > >
> > > Thank you for the quick updates! I really appreciate your responsiveness.
> > >
> > > Regarding large, sparse graphs: Yes, it is tough to find public datasets on the billion-node scale. I hope to release some larger networks (~100 million nodes) later this year, but I don't expect to have the data ready anytime soon. It's also a great point that the powerlaw structure of most real-world networks is likely to help a lot, and I look forward to experimenting with FastGCN-style sampling in some big graphs :)
> > >
> > > Cheers,
> > > Will

---

### Author Response · Authors · 2018-01-16
**Minor revision is done to address a reviewer's comments after the rebuttal period**

- Changed the integral kernel notation \hat{a}(u,v) to \hat{A}(u,v)
- Added training accuracy plots in Fig 4

---

### Author Response · Authors · 2018-11-02
**Announcement of new graph data set**

Since readers landing on this page may be interested in graph deep learning, I, as an author of this work, take the liberty to announce that with another team we compiled a benchmark graph classification data set for public use. The data set is based on problems arising from AI planning. Interested readers are referred to the following website for more details

https://github.com/IBM/IPC-graph-data

Please direct your questions and comments on that website but not here. Thank you.

Jie Chen

---

### Decision · Program_Chairs · 2018-01-29
**ICLR 2018 Conference Acceptance Decision**

**Decision:**

Accept (Poster)

**Comment:**

Graph neural networks (incl. GCNs) have been shown effective on a large range of tasks. However, it has been so far hard (i.e. computationally expensive or requiring the use of heuristics) to apply them to large graphs. This paper aims to address this problem and the solution is clean and elegant. The reviewers generally find it well written and interesting. There were some concerns about the comparison to GraphSAGE (an alternative approach), but these have been addressed in a subsequent revision.

+ an important problem
+ a simple approach
+ convincing results
+ clear and well written